# Multi-View Perceptron: a Deep Model for Learning Face Identity and View Representations

**Zhenyao Zhu**[1,3]     **Ping Luo**[3,1]     **Xiaogang Wang**[2,3]     **Xiaoou Tang**[1,3]

[1]Department of Information Engineering, The Chinese University of Hong Kong
[2]Department of Electronic Engineering, The Chinese University of Hong Kong
[3]Shenzhen Key Lab of CVPR, Shenzhen Institutes of Advanced Technology,
Chinese Academy of Sciences, Shenzhen, China

{zz012,lp011}@ie.cuhk.edu.hk  xgwang@ee.cuhk.edu.hk  xtang@ie.cuhk.edu.hk

## Abstract

Various factors, such as identity, view, and illumination, are coupled in face images. Disentangling the identity and view representations is a major challenge in face recognition. Existing face recognition systems either use handcrafted features or learn features discriminatively to improve recognition accuracy. This is different from the behavior of primate brain. Recent studies [5, 19] discovered that primate brain has a face-processing network, where view and identity are processed by different neurons. Taking into account this instinct, this paper proposes a novel deep neural net, named multi-view perceptron (MVP), which can untangle the identity and view features, and in the meanwhile infer a full spectrum of multi-view images, given a single 2D face image. The identity features of MVP achieve superior performance on the MultiPIE dataset. MVP is also capable to interpolate and predict images under viewpoints that are unobserved in the training data.

## 1   Introduction

The performance of face recognition systems depends heavily on facial representation, which is naturally coupled with many types of face variations, such as view, illumination, and expression. As face images are often observed in different views, a major challenge is to untangle the face identity and view representations. Substantial efforts have been dedicated to extract identity features by hand, such as LBP [1], Gabor [14], and SIFT [15]. The best practise of face recognition extracts the above features on the landmarks of face images with multiple scales and concatenates them into high dimensional feature vectors [4, 21]. Deep learning methods, such as Boltzmann machine [9], sum product network [17], and deep neural net [16, 25, 22, 23, 24, 26] have been applied to face recognition. For instance, Sun et al. [25, 22] employed deep neural net to learn identity features from raw pixels by predicting $10,000$ identities.

Deep neural net is inspired by the understanding of hierarchical cortex in the primate brain and mimicking some aspects of its activities. Recent studies [5, 19] discovered that macaque monkeys have a face-processing network that was made of six interconnected face-selective regions, where neurons in some of these regions were view-specific, while some others were tuned to identity across views, making face recognition in brain of primate robust to view variation. This intriguing function of primate brain inspires us to develop a novel deep neural net, called multi-view perceptron (MVP), which can disentangle identity and view representations, and also reconstruct images under multiple views. Specifically, given a single face image of an identity under an arbitrary view, it can generate a sequence of output face images of the same identity, one at a time, under a full spectrum of viewpoints. Examples of the input images and the generated multi-view outputs of two identities are illustrated in Fig. 1. The images in the last two rows are from the same person. The extracted features of MVP with respect to identity and view are plotted correspondingly in blue and orange.

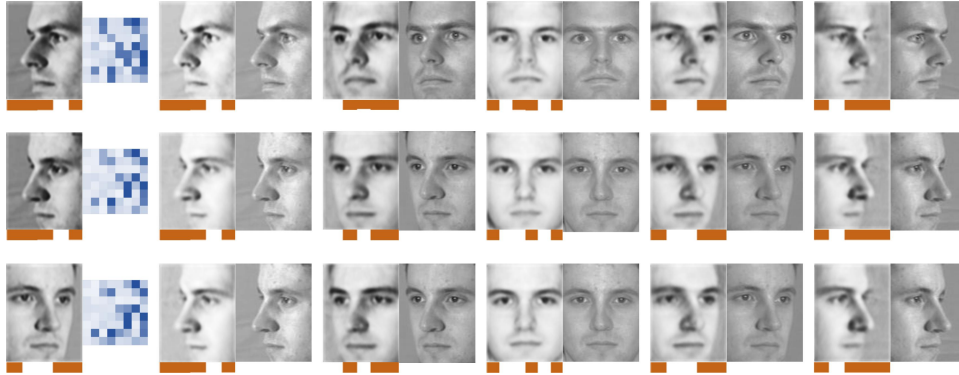

Figure 1: *The inputs (first column) and the multi-view outputs (remaining columns) of two identities. The first input is from one identity and the last two inputs are from the other. Each reconstructed multi-view image (left) has its ground truth (right) for comparison. The extracted identity features of the inputs (the second column), and the view features of both the inputs and outputs are plotted in blue and orange, respectively. The identity features of the same identity are similar, even though the inputs are captured in diverse views, while the view features of the same viewpoint are similar, although they are from different identities. The two persons look similar in the frontal view, but can be better distinguished in other views.*

We can observe that the identity features of the same identity are similar, even though the inputs are captured in very different views, whilst the view features of images in the same view are similar, although they are across different identities.

Unlike other deep networks that produce a deterministic output from an input, MVP employs the deterministic hidden neurons to learn the identity features, whilst using the random hidden neurons to capture the view representation. By sampling distinct values of the random neurons, output images in distinct views are generated. Moreover, to yield images of different viewpoints, we add regularization that images under similar viewpoints should have similar view representations on the random neurons. The two types of neurons are modeled in a probabilistic way. In the training stage, the parameters of MVP are updated by back-propagation, where the gradient is calculated by maximizing a variational lower bound of the complete data log-likelihood. With our proposed learning algorithm, the EM updates on the probabilistic model are converted to forward and backward propagation. In the testing stage, given an input image, MVP can extract its identity and view features. In addition, if an order of viewpoints is also provided, MVP can sequentially reconstruct multiple views of the input image by following this order.

This paper has several key **contributions**. (i) We propose a multi-view perceptron (MVP) and its learning algorithm to factorize the identity and view representations with different sets of neurons, making the learned features more discriminative and robust. (ii) MVP can reconstruct a full spectrum of views given a single 2D image. The full spectrum of views can better distinguish identities, since different identities may look similar in a particular view but differently in others as illustrated in Fig. 1. (iii) MVP can interpolate and predict images under viewpoints that are unobserved in the training data, in some sense imitating the reasoning ability of human.

**Related Works.** In the literature of computer vision, existing methods that deal with view (pose) variation can be divided into 2D- and 3D-based methods. For example, the 2D methods, such as [6], infer the deformation (e.g. thin plate splines) between 2D images across poses. The 3D methods, such as [2, 12], capture 3D face models in different parametric forms. The above methods have their inherent shortages. Extra cost and resources are necessitated to capture and process 3D data. Because of lacking one degree of freedom, inferring 3D deformation from 2D transformation is often ill-posed. More importantly, none of the existing approaches simulates how the primate brain encodes view representations. In our approach, instead of employing any geometric models, view information is encoded with a small number of neurons, which can recover the full spectrum of views together with identity neurons. This representation of encoding identity and view information into different neurons is closer to the face-processing system in the primate brain and new to the deep learning literature. Our previous work [28] learned identity features by using CNN to recover a single frontal view face image, which is a special case of MVP after removing the random neurons. [28] did not learn the view representation as we do. Experimental results show that our approach not only provides rich multi-view representation but also learns better identity features compared with

[28]. Fig. 1 shows examples that different persons may look similar in the front view, but are better distinguished in other views. Thus it improves the performance of face recognition significantly. More recently, Reed et al. [20] untangled factors of image variation by using a high-order Boltzmann machine, where all the neurons are stochastic and it is solved by gibbs sampling. MVP contains both stochastic and deterministic neurons and thus can be efficiently solved by back-propagation.

## 2 Multi-View Perceptron

The training data is a set of image pairs, $\mathcal{I} = \{\mathbf{x}_{ij}, (\mathbf{y}_{ik}, \mathbf{v}_{ik})\}_{i=1,j=1,k=1}^{N,M,M}$, where $\mathbf{x}_{ij}$ is the input image of the $i$-th identity under the $j$-th view, $\mathbf{y}_{ik}$ denotes the output image of the same identity in the $k$-th view, and $\mathbf{v}_{ik}$ is the view label of the output. $\mathbf{v}_{ik}$ is a $M$ dimensional binary vector, with the $k$-th element as 1 and the remaining zeros. MVP is learned from the training data such that given an input $\mathbf{x}$, it can output images $\mathbf{y}$ of the same identity in different views and their view labels $\mathbf{v}$. Then, the output $\mathbf{v}$ and $\mathbf{y}$ are generated as[1],

$$\mathbf{v} = F(\mathbf{y}, \mathbf{h}^v; \Theta), \ \mathbf{y} = F(\mathbf{x}, \mathbf{h}^{id}, \mathbf{h}^v, \mathbf{h}^r; \Theta) + \boldsymbol{\epsilon}, \quad (1)$$

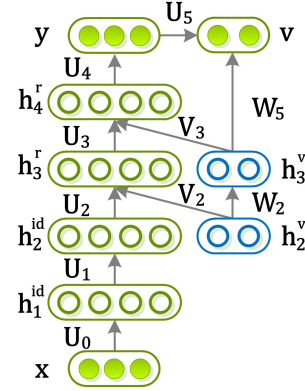

Figure 2: *Network structure of MVP, which has six layers, including three layers with only the deterministic neurons (i.e. the layers parameterized by the weights of $\mathbf{U}_0, \mathbf{U}_1, \mathbf{U}_4$), and three layers with both the deterministic and random neurons (i.e. the weights of $\mathbf{U}_2, \mathbf{V}_2, \mathbf{W}_2, \mathbf{U}_3, \mathbf{V}_3, \mathbf{U}_5, \mathbf{W}_5$). This structure is used throughout the experiments.*

where $F$ is a non-linear function and $\Theta$ is a set of weights and biases to be learned. There are three types of hidden neurons, $\mathbf{h}^{id}$, $\mathbf{h}^v$, and $\mathbf{h}^r$, which respectively extract identity features, view features, and the features to reconstruct the output face image. $\boldsymbol{\epsilon}$ signifies a noise variable.

Fig. 2 shows the architecture[2] of MVP, which is a directed graphical model with six layers, where the nodes with and without filling represent the observed and hidden variables, and the nodes in green and blue indicate the deterministic and random neurons, respectively. The generation process of $\mathbf{y}$ and $\mathbf{v}$ starts from $\mathbf{x}$, flows through the neurons that extract identity feature $\mathbf{h}^{id}$, which combines with the hidden view representation $\mathbf{h}^v$ to yield the feature $\mathbf{h}^r$ for face recovery. Then, $\mathbf{h}^r$ generates $\mathbf{y}$.

Meanwhile, both $\mathbf{h}^v$ and $\mathbf{y}$ are united to generate $\mathbf{v}$. $\mathbf{h}^{id}$ and $\mathbf{h}^r$ are the deterministic binary hidden neurons, while $\mathbf{h}^v$ are random binary hidden neurons sampled from a distribution $q(\mathbf{h}^v)$. Different sampled $\mathbf{h}^v$ generates different $\mathbf{y}$, making the perception of multi-view possible. $\mathbf{h}^v$ usually has a low dimensionality, approximately ten, as ten binary neurons can ideally model $2^{10}$ distinct views.

For clarity of derivation, we take an example of MVP that contains only one hidden layer of $\mathbf{h}^{id}$ and $\mathbf{h}^v$. More layers can be added and derived in a similar fashion. We consider a joint distribution, which marginalizes out the random hidden neurons,

$$p(\mathbf{y}, \mathbf{v} \,|\mathbf{h}^{id}; \Theta) = \sum_{\mathbf{h}^v} p(\mathbf{y}, \mathbf{v}, \mathbf{h}^v | \mathbf{h}^{id}; \Theta) = \sum_{\mathbf{h}^v} p(\mathbf{v} \,|\mathbf{y}, \mathbf{h}^v; \Theta)p(\mathbf{y}|\mathbf{h}^{id}, \mathbf{h}^v; \Theta)p(\mathbf{h}^v), \quad (2)$$

where $\Theta = \{\mathbf{U}_0, \mathbf{U}_1, \mathbf{V}_1, \mathbf{U}_2, \mathbf{V}_2\}$, the identity feature is extracted from the input image, $\mathbf{h}^{id} = f(\mathbf{U}_0\mathbf{x})$, and $f$ is the sigmoid activation function, $f(x) = 1/(1 + \exp(-x))$. Other activation functions, such as rectified linear function [18] and tangent [11], can be used as well. To model continuous values of the output, we assume $\mathbf{y}$ follows a conditional diagonal Gaussian distribution, $p(\mathbf{y}|\mathbf{h}^{id}, \mathbf{h}^v; \Theta) = \mathcal{N}(\mathbf{y}|\mathbf{U}_1\mathbf{h}^{id} + \mathbf{V}_1\mathbf{h}^v, \boldsymbol{\sigma}_\mathbf{y}^2)$. The probability of $\mathbf{y}$ belonging to the $j$-th view is modeled with the softmax function, $p(\mathbf{v}_j = 1|\mathbf{y}, \mathbf{h}^v; \Theta) = \frac{\exp(\mathbf{U}_{j*}^2\mathbf{y} + \mathbf{V}_{j*}^2\mathbf{h}^v)}{\sum_{k=1}^{K} \exp(\mathbf{U}_{k*}^2\mathbf{y} + \mathbf{V}_{k*}^2\mathbf{h}^v)}$, where $\mathbf{U}_{j*}$ indicates the $j$-th row of the matrix.

## 2.1 Learning Procedure

The weights and biases of MVP are learned by maximizing the data log-likelihood. The lower bound of the log-likelihood can be written as,

$$\log p(\mathbf{y}, \mathbf{v} \,|\mathbf{h}^{id}; \Theta) = \log \sum_{\mathbf{h}^v} p(\mathbf{y}, \mathbf{v}, \mathbf{h}^v | \mathbf{h}^{id}; \Theta) \geq \sum_{\mathbf{h}^v} q(\mathbf{h}^v) \log \frac{p(\mathbf{y}, \mathbf{v}, \mathbf{h}^v | \mathbf{h}^{id}; \Theta)}{q(\mathbf{h}^v)}. \quad (3)$$

Eq.(3) is attained by decomposing the log-likelihood into two terms, $\log p(\mathbf{y}, \mathbf{v} \,|\mathbf{h}^{id}; \Theta) = -\sum_{\mathbf{h}^v} q(\mathbf{h}^v) \log \frac{p(\mathbf{h}^v | \mathbf{y}, \mathbf{v}; \Theta)}{q(\mathbf{h}^v)} + \sum_{\mathbf{h}^v} q(\mathbf{h}^v) \log \frac{p(\mathbf{y}, \mathbf{v}, \mathbf{h}^v | \mathbf{h}^{id}; \Theta)}{q(\mathbf{h}^v)}$, which can be easily verified by substituting the product, $p(\mathbf{y}, \mathbf{v}, \mathbf{h}^v | \mathbf{h}^{id}) = p(\mathbf{y}, \mathbf{v} \,|\mathbf{h}^{id}) p(\mathbf{h}^v | \mathbf{y}, \mathbf{v})$, into the right hand side of the decomposition. In particular, the first term is the KL-divergence [10] between the true posterior and the distribution $q(\mathbf{h}^v)$. As KL-divergence is non-negative, the second term is regarded as the variational lower bound on the log-likelihood.

The above lower bound can be maximized by using the Monte Carlo Expectation Maximization (MCEM) algorithm recently introduced by [27], which approximates the true posterior by using the importance sampling with the conditional prior as the proposal distribution. With the Bayes' rule, the true posterior of MVP is $p(\mathbf{h}^v | \mathbf{y}, \mathbf{v}) = \frac{p(\mathbf{y}, \mathbf{v} \,|\mathbf{h}^v) p(\mathbf{h}^v)}{p(\mathbf{y}, \mathbf{v})}$, where $p(\mathbf{y}, \mathbf{v} \,|\mathbf{h}^v)$ represents the multi-view perception error, $p(\mathbf{h}^v)$ is the prior distribution over $\mathbf{h}^v$, and $p(\mathbf{y}, \mathbf{v})$ is a normalization constant. Since we do not assume any prior information on the view distribution, $p(\mathbf{h}^v)$ is chosen as a uniform distribution between zero and one. To estimate the true posterior, we let $q(\mathbf{h}^v) = p(\mathbf{h}^v | \mathbf{y}, \mathbf{v}; \Theta^{old})$. It is approximated by sampling $\mathbf{h}^v$ from the uniform distribution, i.e. $\mathbf{h}^v \sim \mathcal{U}(0,1)$, weighted by the importance weight $p(\mathbf{y}, \mathbf{v} \,|\mathbf{h}^v; \Theta^{old})$. With the EM algorithm, the lower bound of the log-likelihood turns into

$$\mathcal{L}(\Theta, \Theta^{old}) = \sum_{\mathbf{h}^v} p(\mathbf{h}^v | \mathbf{y}, \mathbf{v}; \Theta^{old}) \log p(\mathbf{y}, \mathbf{v}, \mathbf{h}^v | \mathbf{h}^{id}; \Theta) \simeq \frac{1}{S} \sum_{s=1}^{S} w_s \log p(\mathbf{y}, \mathbf{v}, \mathbf{h}_s^v | \mathbf{h}^{id}; \Theta),$$

$$(4)$$

where $w_s = p(\mathbf{y}, \mathbf{v} \,|\mathbf{h}^v; \Theta^{old})$ is the importance weight. The E-step samples the random hidden neurons, i.e. $\mathbf{h}_s^v \sim \mathcal{U}(0,1)$, while the M-step calculates the gradient,

$$\frac{\partial \mathcal{L}}{\partial \Theta} \simeq \frac{1}{S} \sum_{s=1}^{S} \frac{\partial \mathcal{L}(\Theta, \Theta^{old})}{\partial \Theta} = \frac{1}{S} \sum_{s=1}^{S} w_s \frac{\partial}{\partial \Theta} \{\log p(\mathbf{v} \,|\mathbf{y}, \mathbf{h}_s^v) + \log p(\mathbf{y}|\mathbf{h}^{id}, \mathbf{h}_s^v)\}, \quad (5)$$

where the gradient is computed by averaging over all the gradients with respect to the importance samples.

The two steps have to be iterated. When more samples are needed to estimate the posterior, the space complexity will increase significantly, because we need to store a batch of data, the proposed samples, and their corresponding outputs at each layer of the deep network. When implementing the algorithm with GPU, one needs to make a tradeoff between the size of the data and the accurateness of the approximation, if the GPU memory is not sufficient for large scale training data. Our empirical study (Sec. 3.1) shows that the M-step of MVP can be computed by using only one sample, because the uniform prior typically leads to sparse weights during training. Therefore, the EM process develops into the conventional back-propagation.

In the **forward** pass, we sample a number of $\mathbf{h}_s^v$ based on the current parameters $\Theta$, such that only the sample with the largest weight need to be stored. We demonstrate in the experiment (Sec. 3.1) that a small number of times (e.g. $< 20$) are sufficient to find good proposal. In the **backward** pass, we seek to update the parameters by the gradient,

$$\frac{\partial \mathcal{L}(\Theta)}{\partial \Theta} \simeq \frac{\partial}{\partial \Theta} \{w_s \big(\log p(\mathbf{v} \,|\mathbf{y}, \mathbf{h}_s^v) + \log p(\mathbf{y}|\mathbf{h}^{id}, \mathbf{h}_s^v)\big)\}, \quad (6)$$

where $\mathbf{h}_s^v$ is the sample that has the largest weight $w_s$. We need to optimize the following two terms, $\log p(\mathbf{y}|\mathbf{h}^{id}, \mathbf{h}_s^v) = -\log \boldsymbol{\sigma}_{\mathbf{y}} - \frac{\|\widehat{\mathbf{y}} - (\mathbf{U}_1 \mathbf{h}^{id} + \mathbf{V}_1 \mathbf{h}_s^v)\|_2^2}{2\boldsymbol{\sigma}_{\mathbf{y}}^2}$ and $\log p(\mathbf{v} \,|\mathbf{y}, \mathbf{h}_s^v) = \sum_j \widehat{\mathbf{v}}_j \log\big(\frac{\exp(\mathbf{U}_{j*}^2 \mathbf{y} + \mathbf{V}_{j*}^2 \mathbf{h}_s^v)}{\sum_{k=1}^{K} \exp(\mathbf{U}_{k*}^2 \mathbf{y} + \mathbf{V}_{k*}^2 \mathbf{h}_s^v)}\big)$, where $\widehat{\mathbf{y}}$ and $\widehat{\mathbf{v}}$ are the ground truth.

● **Continuous View** In the previous discussion, $\mathbf{v}$ is assumed to be a binary vector. Note that $\mathbf{v}$ can also be modeled as a continuous variable with a Gaussian distribution,

$$p(\mathbf{v}\,|\mathbf{y},\mathbf{h}^v) = \mathcal{N}(\mathbf{v}\,|\mathbf{U}_2\mathbf{y} + \mathbf{V}_2\mathbf{h}^v, \boldsymbol{\sigma}_\mathbf{v}), \tag{7}$$

where $\mathbf{v}$ is a scalar corresponding to different views from $-90°$ to $+90°$. In this case, we can generate views not presented in the training data by interpolating $\mathbf{v}$, as shown in Fig. 6.

● **Difference with multi-task learning** Our model, which only has a single task, is also different from multi-task learning (MTL), where reconstruction of each view could be treated as a different task, although MTL has not been used for multi-view reconstruction in literature to the best of our knowledge. In MTL, the number of views to be reconstructed is predefined, equivalent to the number of tasks, and it encounters problems when the training data of different views are unbalanced; while our approach can sample views continuously and generate views not presented in the training data by interpolating $\mathbf{v}$ as described above. Moreover, the model complexity of MTL increases as the number of views and its training is more difficult since different tasks may have difference convergence rates.

## 2.2 Testing Procedure

Given the view label $\mathbf{v}$, and the input $\mathbf{x}$, we generate the face image $\mathbf{y}$ under the viewpoint of $\mathbf{v}$ in the testing stage. A set of $\mathbf{h}^v$ are first sampled, $\{\mathbf{h}^v_s\}^S_{s=1} \sim \mathcal{U}(0,1)$, which corresponds to a set of outputs $\{\mathbf{y}_s\}^S_{s=1}$. For example, in a simple network with only one hidden layer, $\mathbf{y}_s = \mathbf{U}_1\mathbf{h}^{id} + \mathbf{V}_1\mathbf{h}^v_s$ and $\mathbf{h}^{id} = f(\mathbf{U}_0\mathbf{x})$. Then, the desired face image in view $\mathbf{v}$ is the output $\mathbf{y}_s$ that produces the largest probability of $p(\mathbf{v}\,|\mathbf{y}_s,\mathbf{h}^v_s)$. A full spectrum of multi-view images are reconstructed for all the possible view labels $\mathbf{v}$.

## 2.3 View Estimation

Our model can also be used to estimate viewpoint of the input image $\mathbf{x}$. First, given all possible values of viewpoint $\mathbf{v}$, we can generate a set of corresponding output images $\{\mathbf{y}_z\}$, where $z$ indicates the index of the values of view we generated (or interpolated). Then, to estimate viewpoint, we assign the view label of the $z$-th output $\mathbf{y}_z$ to $\mathbf{x}$, such that $\mathbf{y}_z$ is the most similar image to $\mathbf{x}$. The above procedure is formulated as below. If $\mathbf{v}$ is discrete, the problem is, $\arg\min_{j,z}\parallel p(\mathbf{v}_j = 1|\mathbf{x},\mathbf{h}^v_z) - p(\mathbf{v}_j = 1|\mathbf{y}_z,\mathbf{h}^v_z)\parallel^2_2 = \arg\min_{j,z}\parallel \frac{\exp(\mathbf{U}^2_{j*}\mathbf{x}+\mathbf{V}^2_{j*}\mathbf{h}^v_z)}{\sum^K_{k=1}\exp(\mathbf{U}^2_{k*}\mathbf{x}+\mathbf{V}^2_{k*}\mathbf{h}^v_z)}$ $-\frac{\exp(\mathbf{U}^2_{j*}\mathbf{y}_z+\mathbf{V}^2_{j*}\mathbf{h}^v_z)}{\sum^K_{k=1}\exp(\mathbf{U}^2_{k*}\mathbf{y}_z+\mathbf{V}^2_{k*}\mathbf{h}^v_z)}\parallel^2_2$. If $\mathbf{v}$ is continuous, the problem is defined as, $\arg\min_z\parallel (\mathbf{U}_2\mathbf{x} + \mathbf{V}_2\mathbf{h}^v_z) - (\mathbf{U}_2\mathbf{y}_z + \mathbf{V}_2\mathbf{h}^v_z)\parallel^2_2 = \arg\min_z\parallel \mathbf{x} - \mathbf{y}_z\parallel^2_2$.

# 3 Experiments

Several experiments are designed for evaluation and comparison[3]. In Sec. 3.1, MVP is evaluated on a large face recognition dataset to demonstrate the effectiveness of the identity representation. Sec. 3.2 presents a quantitative evaluation, showing that the reconstructed face images are in good quality and the multi-view spectrum has retained discriminative information for face recognition. Sec. 3.3 shows that MVP can be used for view estimation and achieves comparable result as the discriminative methods specially designed for this task. An interesting experiment in Sec. 3.4 shows that by modeling the view as a continuous variable, MVP can analyze and reconstruct views not seen in the training data.

## 3.1 Multi-View Face Recognition

MVP on multi-view face recognition is evaluated on the MultiPIE dataset [7], which contains $754,204$ images of $337$ identities. Each identity was captured under $15$ viewpoints from $-90°$ to $+90°$ and $20$ different illuminations. It is the largest and most challenging dataset for evaluating face recognition under view and lighting variations. We conduct the following three experiments to demonstrate the effectiveness of MVP.

• **Face recognition across views** This setting follows the existing methods, e.g. [2, 12, 28], which employs the same subset of MultiPIE that covers images from $-45°$ to $+45°$ and with neutral illumination. The first 200 identities are used for training and the remaining 137 identities for test. In the testing stage, the gallery is constructed by choosing one canonical view image $(0°)$ from each testing identity. The remaining images of the testing identities from $-45°$ to $+45°$ are selected as probes. The number of neurons in MVP can be expressed as $32 \times 32 - 512 - 512(10) - 512(10) - 1024 - 32 \times 32[7]$, where the input and output images have the size of $32 \times 32$, [7] denotes the length of the view label vector ($\mathbf{v}$), and $(10)$ represents that the third and forth layers have ten random neurons.

We examine the performance of using the identity features, i.e. $\mathbf{h}_2^{id}$ (denoted as $\mathrm{MVP}_{\mathbf{h}_2^{id}}$), and compare it with seven state-of-the-art methods in Table 1. The first three methods are based on 3D face models and the remaining ones are 2D feature extraction methods, including deep models, such as FIP [28] and RL [28], which employed the traditional convolutional network to recover the frontal view face image. As the existing methods did, LDA is applied to all the 2D methods to reduce the features' dimension. The first and the second best results are highlighted for each viewpoint, as shown in Table 1. The two deep models (MVP and RL) outperform all the existing methods, including the 3D face models. RL achieves the best results on three viewpoints, whilst MVP is the best on four viewpoints. The extracted feature dimensions of MVP and RL are 512 and 9216, respectively. In summary, MVP obtains comparable averaged accuracy as RL under this setting, while the learned feature representation is more compact.

Table 1: *Face recognition accuracies across views. The first and the second best performances are in bold.*

|  | Avg. | $-15°$ | $+15°$ | $-30°$ | $+30°$ | $-45°$ | $+45°$ |
|---|---|---|---|---|---|---|---|
| VAAM [2] | 86.9 | 95.7 | 95.7 | 89.5 | 91.0 | 74.1 | 74.8 |
| FA-EGFC [12] | 92.7 | 99.3 | 99.0 | 92.9 | 95.0 | 84.7 | 85.2 |
| SA-EGFC [12] | 97.2 | 99.7 | 98.3 | **99.7** | **98.7** | 93.0 | 93.6 |
| LE [3]+LDA | 93.2 | 99.9 | **99.7** | 95.5 | 95.5 | 86.9 | 81.8 |
| CRBM [9]+LDA | 87.6 | 94.9 | 96.4 | 88.3 | 90.5 | 80.3 | 75.2 |
| FIP [28]+LDA | 95.6 | **100.0** | 98.5 | 96.4 | 95.6 | 93.4 | 89.8 |
| RL [28]+LDA | **98.3** | **100.0** | 99.3 | 98.5 | 98.5 | **95.6** | **97.8** |
| $\mathrm{MVP}_{\mathbf{h}_2^{id}}$+LDA | **98.1** | **100.0** | **100.0** | **100.0** | **99.3** | 93.4 | 95.6 |

Table 2: *Face recognition accuracies across views and illuminations. The first and the second best performances are in bold.*

|  | Avg. | $0°$ | $-15°$ | $+15°$ | $-30°$ | $+30°$ | $-45°$ | $+45°$ | $-60°$ | $+60°$ |
|---|---|---|---|---|---|---|---|---|---|---|
| Raw Pixels+LDA | 36.7 | 81.3 | 59.2 | 58.3 | 35.5 | 37.3 | 21.0 | 19.7 | 12.8 | 7.63 |
| LBP [1]+LDA | 50.2 | 89.1 | 77.4 | 79.1 | 56.8 | 55.9 | 35.2 | 29.7 | 16.2 | 14.6 |
| Landmark LBP [4]+LDA | 63.2 | 94.9 | 83.9 | 82.9 | 71.4 | 68.2 | 52.8 | 48.3 | 35.5 | 32.1 |
| CNN+LDA | 58.1 | 64.6 | 66.2 | 62.8 | 60.7 | 63.6 | 56.4 | 57.9 | 46.4 | 44.2 |
| FIP [28]+LDA | 72.9 | 94.3 | 91.4 | 90.0 | 78.9 | 82.5 | 66.1 | 62.0 | 49.3 | 42.5 |
| RL [28]+LDA | 70.8 | 94.3 | 90.5 | 89.8 | 77.5 | 80.0 | 63.6 | 59.5 | 44.6 | 38.9 |
| MTL+RL+LDA | **74.8** | **93.8** | **91.7** | **89.6** | 80.1 | 83.3 | 70.4 | **63.8** | 51.5 | 50.2 |
| $\mathrm{MVP}_{\mathbf{h}_1^{id}}$+LDA | 61.5 | 92.5 | 85.4 | 84.9 | 64.3 | 67.0 | 51.6 | 45.4 | 35.1 | 28.3 |
| $\mathrm{MVP}_{\mathbf{h}_2^{id}}$+LDA | **79.3** | **95.7** | **93.3** | **92.2** | **83.4** | **83.9** | **75.2** | **70.6** | **60.2** | **60.0** |
| $\mathrm{MVP}_{\mathbf{h}_3^{r}}$+LDA | 72.6 | 91.0 | 86.7 | 84.1 | 74.6 | 74.2 | 68.5 | **63.8** | **55.7** | **56.0** |
| $\mathrm{MVP}_{\mathbf{h}_4^{r}}$+LDA | 62.3 | 83.4 | 77.3 | 73.1 | 62.0 | 63.9 | 57.3 | 53.2 | 44.4 | 46.9 |

• **Face recognition across views and illuminations** To examine the robustness of different feature representations under more challenging conditions, we extend the first setting by employing a larger subset of MultiPIE, which contains images from $-60°$ to $+60°$ and 20 illuminations. Other experimental settings are the same as the above. In Table 2, feature representations of different layers in MVP are compared with seven existing features, including raw pixels, LBP [1] on image grid, LBP on facial landmarks [4], CNN features, FIP [28], RL [28], and MTL+RL. LDA is applied to all the feature representations. Note that the last four methods are built on the convolutional neural networks. The only distinction is that they adopted different objective functions to learn features. Specifically, CNN uses cross-entropy loss to classify face identity as in [26]. FIP and RL utilized least-square loss to recover the frontal view image. MTL+RL is an extension of RL. It employs multiple tasks, each of which is formulated as a least square loss, to recover multi-view images, and all the tasks share feature layers. To achieve fair comparisons, CNN, FIP, and MTL+RL adopt the same convolutional structure as RL [28], since RL achieves competitive results in our first experiment.

The first and second best results are emphasized in bold in Table 2. The identity feature $\mathbf{h}_2^{id}$ of MVP outperforms all the other methods on all the views with large margins. MTL+RL achieves the second best results except on $\pm 60^\circ$. These results demonstrate the superior of modeling multi-view perception. For the features at different layers of MVP, the performance can be summarized as $\mathbf{h}_2^{id} > \mathbf{h}_3^r > \mathbf{h}_1^{id} > \mathbf{h}_4^r$, which conforms our expectation. $\mathbf{h}_2^{id}$ performs the best because it is the highest level of identity features. $\mathbf{h}_2^{id}$ performs better than $\mathbf{h}_1^{id}$ because pose factors coupled in the input image $\mathbf{x}$ have be further removed, after one more forward mapping from $\mathbf{h}_1^{id}$ to $\mathbf{h}_2^{id}$. $\mathbf{h}_2^{id}$ also outperforms $\mathbf{h}_3^r$ and $\mathbf{h}_4^r$, because some randomly generated view factors ($\mathbf{h}_2^v$ and $\mathbf{h}_3^v$) have been incorporated into these two layers during the construction of the full view spectrum. Please refer to Fig. 2 for a better understanding.

• **Effectiveness of the BP Procedure**

Fig. 3 (a) compares the convergence rates during training, when using different number of samples to estimate the true posterior. We observe that a few number of samples, such as twenty, can lead to reasonably good convergence. Fig. 3 (b) empirically shows that uniform prior leads to sparse weights during training. In other words,

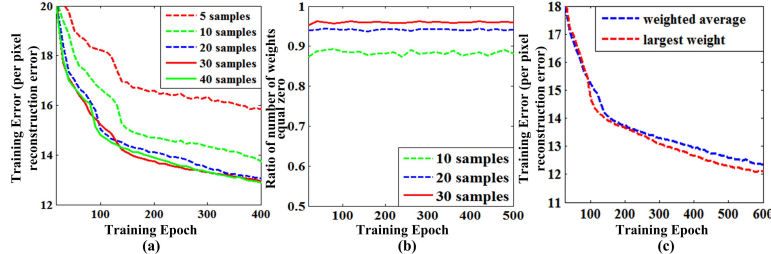

Figure 3: *Analysis of MVP on the MultiPIE dataset. (a) Comparison of convergence, using different number of samples to estimate the true posterior. (b) Comparison of sparsity of the samples' weights. (c) Comparison of convergence, using the largest weighted sample and using the weighted average over all the samples to compute gradient.*

if we seek to calculate the gradient of BP using only one sample, as did in Eq.(6). Fig. 3 (b) demonstrates that 20 samples are sufficient, since only 6 percent of the samples' weights approximate one (all the others are zeros). Furthermore, as shown in Fig. 3 (c), the convergence rates of the one-sample gradient and the weighted summation are comparable.

## 3.2 Reconstruction Quality

Another experiment is designed to quantitatively evaluate the multi-view reconstruction result. The setting is the same as the first experiment in Sec. 3.1. The gallery images are all in the frontal view ($0^\circ$). Differently, LDA is applied to the raw pixels of the original images (OI) and the reconstructed images (RI) under the same view, respectively. Fig. 4 plots the accuracies of face recognition with respect to distinct viewpoints. Not surprisingly, under the viewpoints of $+30^\circ$ and $-45^\circ$ the accuracies of RI are decreased compared to OI. Nevertheless, this decrease is comparatively small ($< 5\%$). It implies that the reconstructed images are in reasonably good quality. We notice that the reconstructed images in Fig. 1 lose some detailed textures, while well preserving the shapes of profile and the facial components.

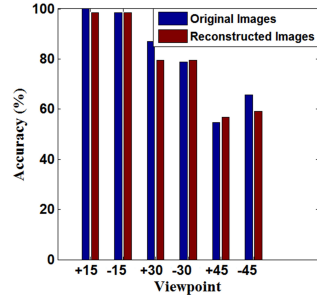

Figure 4: *Face recognition accuracies. LDA is applied to the raw pixels of the original images and the reconstructed images.*

## 3.3 Viewpoint Estimation

This experiment is conducted to evaluate the performance of viewpoint estimation. MVP is compared to Linear Regression (LR) and Support Vector Regression (SVR), both of which have been used in viewpoint estimation, e.g. [8, 13]. Similarly, we employ the first setting as introduced in Sec. 3.1, implying that we train the models using images of a set of identities, and then estimate poses of the images of the remaining identities. For training LR and SVR, the features are obtained by applying PCA on the raw image pixels. Fig. 5 reports the view estimation errors, which are measured by the differences between the pose degrees

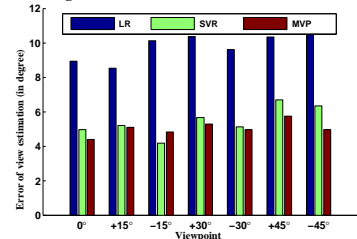

Figure 5: *Errors of view estimation.*

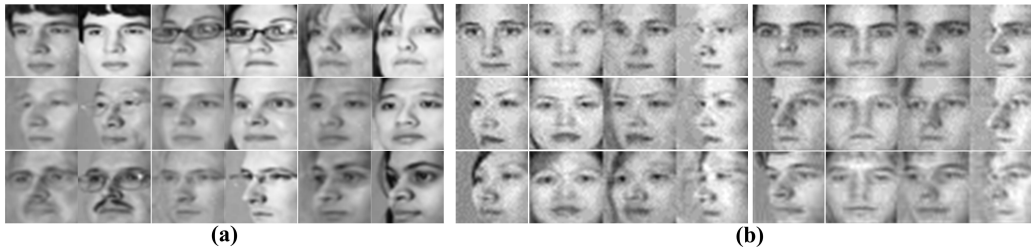

|     (a)     |     (b)     |

Figure 6: *We adopt the images in* $0°$, $30°$, *and* $60°$ *for training, and test whether MVP can analyze and reconstruct images under* $15°$ *and* $45°$. *The reconstructed images (left) and the ground truths (right) are shown in (a). (b) visualizes the full spectrum of the reconstructed images, when the images in unobserved views are used as inputs (first column).*

of ground truth and the predicted degrees. The averaged errors of MVP, LR, and SVR are $5.03°$, $9.79°$, and $5.45°$, respectively. MVP achieves slightly better results compared to the discriminative model, i.e. SVR, demonstrating that it is also capable for view estimation, even though it is not designated for this task.

### 3.4 Viewpoint Interpolation

When the viewpoint is modeled as a continuous variable as described in Sec. 2.1, MVP implicitly captures a 3D face model, such that it can analyze and reconstruct images under viewpoints that have not been seen before, while this cannot be achieved with MTL. In order to verify such capability, we conduct two tests. First, we adopt the images from MultiPIE in $0°$, $30°$, and $60°$ for training, and test whether MVP can generate images under $15°$ and $45°$. For each testing identity, the result is obtained by using the image in $0°$ as input and reconstructing images in $15°$ and $45°$. Several synthesized images (left) compared with the ground truth (right) are visualized in Fig. 6 (a). Although the interpolated images have noise and blurring effect, they have similar views as the ground truth and more importantly, the identity information is preserved. Second, under the same training setting as above, we further examine, when the images of the testing identities in $15°$ and $45°$ are employed as inputs, whether MVP can still generate a full spectrum of multi-view images and preserve identity information in the meanwhile. The results are illustrated in Fig. 6 (b), where the first image is the input and the remaining are the reconstructed images in $0°$, $30°$, and $60°$.

These two experiments show that MVP essentially models a continuous space of multi-view images such that first, it can predict images in unobserved views, and second, given an image under an unseen viewpoint, it can correctly extract identity information and then produce a full spectrum of multi-view images. In some sense, it performs multi-view reasoning, which is an intriguing function of human brain.

## 4 Conclusions

In this paper, we have presented a generative deep network, called Multi-View Perceptron (MVP), to mimic the ability of multi-view perception in primate brain. MVP can disentangle the identity and view representations from an input image, and also can generate a full spectrum of views of the input image. Experiments demonstrated that the identity features of MVP achieve better performance on face recognition compared to state-of-the-art methods. We also showed that modeling the view factor as a continuous variable enables MVP to interpolate and predict images under the viewpoints, which are not observed in training data, imitating the reasoning capacity of human.

**Acknowledgement** This work is partly supported by Natural Science Foundation of China (91320101, 61472410), Shenzhen Basic Research Program (JCYJ20120903092050890, JCYJ20120617114614438, J-CYJ20130402113127496), Guangdong Innovative Research Team Program (201001D0104648280).

## Footnotes

[1]The subscripts $i, j, k$ are omitted for clearness.

[2]For clarity, the biases are omitted.

[3] http://mmlab.ie.cuhk.edu.hk/projects/MVP.htm. For more technical details of this work, please contact the corresponding author Ping Luo (pluo.lhi@gmail.com).

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
