[Reviews · NeurIPS 2014]

Submitted by Assigned_Reviewer_13

This paper presents a novel way to do multi-view representation of faces using a neural-net like model with deterministic and probabilistic units. The network is trained to recover the identity and view. It can be used to synthesize a new view as well. Generally, the paper does a good job at disentangling the contributing factors to the success of the method, and the results seem quite good.

I wish the authors presented more evidence for why they chose to design their model they way they did. For one, it is not clear that a non-deterministic approach is really needed in this case -- could they have done a model that is fully feed-forward with deterministic neurons? It would have certainly avoided the need to variational/importance sampling inference.

Quality: generally, a good paper with an interesting and well-grounded model, evaluated well on a competitive benchmark and

Clarity: The paper is well-written and easy to read, for most of the part.

Originality: in some sense, the approach is comparable to the work of Geoff Hinton and collaborators, who use stacks of Restricted Boltzmann Machines, where the top RBM can have the label as a side

Significance: this is likely to be of interest to the facial recognition community, but also to other researchers who work on problems where a hybrid deterministic and generative approach may be a good solution.

Comments and questions:

- How did the authors come up with the structure in Figure 2? What are the insights behind the design choices made?
- The authors should define q(h) at line 165
- Is the MCEM method crucial to the model presented? More analysis on why this particular optimization method was used is in order since MCEM is not exactly a widely used method in the community.
- For inference: how big is S at line 235? How expensive is inference generally? Some numbers and O() notation results would be good.
- Is LDA necessary to get the best results? How do the results look like without LDA?
- I don't think the authors should be using the tangent with how the brain works, except very sparsely. There is not much evidence that the presented method is particularly close to how humans do facial recognition or view point interpolation.
- Did the authors try their approach on the Labeled Faces in the Wild benchmark? If yes, do they have results to report?
Summary: An interesting novel approach to solving the face recognition problem using a graphical neural net-like model that is trained to recover the identity of the input face and its view. Competitive results on the MultiPIE benchmark and interesting solid experiments.

Submitted by Assigned_Reviewer_33

This paper proposes to use an Autoencoder with both deterministic and stochastic units to learn to reconstruct faces of the same person under different 3D viewpoints.
It is well written and the experiments are pretty solid.

The model in Fig. 2 have certain resemblance to the one in [25]. The main difference is that the v is added and it is predicted by both y and h3. This is useful to be able to use feedforward prediction of v instead of using Bayes rule. It is perhaps a good idea to discuss these differences in the text.

minor:
abstract: 'instinct' could be changed to 'intuition' or 'insight'

in figures 1 and 6 are the examples test or training images?

To make the paper stronger, the authors can try to use the STOA Siamese CNN approach of [24] and see how it compares to the proposed methods.
Summary: This paper proposes to use an autoencoder for pose representation learning on faces data. The problems are interesting and the experiments seems to suggest the usefulness of the approach.

Submitted by Assigned_Reviewer_44

The paper studies multi-view face recognition in constrained environments. The paper proposes a multi-layer multi-view perceptron (MVP) network to infer the identity and view angle representations by deterministic and random neurons from a single 2D image. The MVP is able to synthesize new view of faces from a single 2D face by a sequence of view representations. The paper derives the Monte Carlo Expectation Maximization (MCEM) procedure to learn the MVP from a pair of faces plus the view label, assuming the output face follows a conditional diagonal Gaussian distribution. The MVP is learned from 200 persons in the MultiPIE dataset and the identity features are employed to classify the reset 137 persons by LDA, which achieves comparable performance as the recent methods using convolutional neural networks.

The paper proposes a novel MVP network to disentangle the face identity and view angle representations, and employs the identity features in LDA for multi-view face recognition. Varying the view angle feature in MVP can synthesize new face views. Overall, the technical approach is sound with some limitations and the performance on the MultiPie is very competitive. Some technical concerns are as follows.

The approach seems to depend on well-aligned normalized 32X32 face images. The experiments on face recognition among 137 persons on MultiPIE leave the scalability of the proposed method in question. It is unclear if the method can benefit from more training samples and how to extend the approach for unconstrained face recognition in real applications.

How critical to assume the conditional diagonal Gaussian distribution for the output y? How many training pairs from 200 persons in MultiPIE are used to learn the MVP? Why the new face views have to be "sequentially" synthesized?

The claim that this new MVP "simulates/mimics how human brain encodes view representations" is quite weak. The paper presents no convincing evidence or discussion on how human brain encodes view representations and why the mechanism is similar to MVP. So I would not list this as one key contribution of the paper.
Summary: This is a descent work on multi-view perceptron (MVP) learning for multi-view face recognition. The performance on the MultiPIE is competitive. Though I have the concerns on the scalability of this method and how to apply it to a face recognition system.
Author Feedback
Author rebuttal: To R13
Q1: Could they done a model that is fully feedforward with deterministic neurons
A1: In general, deterministic neurons (e.g. CNN) are good at extracting discriminative features for classification, while stochastic neurons (e.g. DBN) better reconstruct samples. Our net needs to accomplish both tasks, i.e. extracting identity features for face recognition and reconstructing images. So a natural idea is to combine both types of neurons.
In our model, both the reconstructed image y and its view v are output, so the view representation neurons hv have to be treated as random hidden variables and inferred from output v. Benefits are: 1) being able to reconstruct a continuous view spectrum as explained in line 216-221; 2) better modeling the connection between y and v through U_5, pointing from y to v, i.e. viewpoint is estimated from reconstructed image.
A natural implementation with deterministic neurons only is to integrate multitask learning (MTL) and [28]. Its limitation was discussion in line 223-230. Experimental results are shown in Tbl.2. Its face recognition accuracy drops by 5% on average compared with ours. It can't reconstruct continuous and unseen views as we did in Fig.6. An alternative is to treat v as input. It is a mixture model (v selects mixture component), and is like MTL (v selects the task). The produced result is similar to MTL. It loses the direction connection from y to v and can't learn the continuous view space well.

Q2: The approach is comparable to stacks of RBMs
A2: In stacked RBMs, all hidden neurons are stochastic, optimized by Gibbs sampling. We employ both deterministic and stochastic neurons, optimized by BP and EM.

Q3: Is the MCEM method crucial to the model presented
A3: Yes. EM is typically suitable for random neurons in our model. However, the true posterior is difficult to estimate in our case. It is crucial to apply Monte Carlo method. Other reason is MCEM is more efficient than Gibbs sampling and variational inference with mean-field approximation.

Q4: How expensive is inference
A4: S in test stage is chosen the same as in training (S=20). It takes 6.3ms per image with GPU.

Q5: How the results look like without LDA
A5: With euclidean distance, the accuracies of all methods decrease ~10% and MVP is still the best.

Q6: I don't think the authors should be using the tangent with how the brain works, except very sparsely
A6: Agree. We will tone down and put it in the discussion/future work section, or completely remove it. It doesn't hurt the novelty and effectiveness of the proposed method.

Q7: Did the authors try their approach on LFW
A7: We tried to recover frontal view image and then trained CNNs on the recovered image for face verification. We trained on 80K web face images. The accuracy on LFW is 97.20%, which is comparable to [24] 97.35%. However, the size of our training data is much smaller than [24], i.e. 80k v.s. 4 millions. We use MVP (2D model) to normalize images, while [24] used precise landmarks and 3D reconstruction. This result shows that MVP has many potentials to be explored. e.g. 1) with only frontal view, we achieve good performance, not to mention MVP is designed to recover multiview, which is more discriminative; 2) many existing face verification methods can be applied to our recovered images in all views.
However, we target on pose and thus chose MultiPIE instead of LFW for evaluation. Pose variation MultiPIE is much larger than LFW. LFW is a mix of many factors and not suitable to evaluate pose.

To R33
Q1: Differences with [25]
A1: Agree. v makes feedforward prediction possible. v also enables the learning of identity and view representations by deterministic and random neurons, as in A1 to R13. Will add more discussions.

Q2: In fig. 1 and 6 are the examples test or training images?
A2: Test images.

Q3: To make the paper strong, try to use the STOA siamese CNN approach of [24]
A3: Agree. [24] is a general face recognition approach while we focus on pose variation. [24] replies on a 3D face model to align pose variation (see A7 to R13), while we learn 2D models. In this sense, our model is integratable with [24] as a pose normalization step.

To R44
Q1: The approach seems to depend on well-aligned normalized images
A1: Face images were coarsely aligned with only 3 points (eye centers,mouth center)

Q2: unclear if the method can benefit from more training samples and how to extend the approach for unconstrained face recognition
A2: The model can benefit from more training samples and more training people because of the nature deep learning. See A7 to R13. Yet, face recognition depends on multiple factors. Pose is one of them. Our method can be treated as a pose normalization step to be combined with other methods e.g. [24] to further address face recognition in the wild. See A3 to R33.
Web images collected from the wild is not the only scenario of face recognition. In many applications, such as entrance/customs security and surveillance, images are captured more like under controlled environment than LFW.

Q3: How critical to assume the conditional diagonal Gaussian distribution for the output y
A3: Neurons with real values are commonly modeled by diagonal Gaussian, because of its effectiveness and small number of parameters. Other distribution can be adopted, may introduce more parameters.

Q4: How many pairs from 200 persons are used to learn MVP
A4: Each subject has 7 images in 7 poses. So the number of training pairs is 7*7*200.

Q5: Why the new face views have to be 'sequentially' synthesized
A5: 'sequentially' means we predict different viewpoint of the input image one at a time, not multiple views simultaneously. This is to distinguish with multitask learning. The synthesized face images do not need to be in specific order.

Q6: 'simulates/mimics how human brain encodes view representations' is quite weak.
A6: Agree. We could remove it. See A6 to R13.